# Mucus Release and Airway Constriction by TMEM16A May Worsen Pathology in Inflammatory Lung Disease

**DOI:** 10.3390/ijms22157852

**Published:** 2021-07-22

**Authors:** Raquel Centeio, Jiraporn Ousingsawat, Inês Cabrita, Rainer Schreiber, Khaoula Talbi, Roberta Benedetto, Tereza Doušová, Eric K. Verbeken, Kris De Boeck, Isaac Cohen, Karl Kunzelmann

**Affiliations:** 1Physiological Institute, University of Regensburg, Germany University Street 31, D-93053 Regensburg, Germany; raquel.centeio@vkl.uni-regensburg.de (R.C.); jiraporn.ousingsawat@vkl.uni-regensburg.de (J.O.); ines.cabrita@vkl.uni-regensburg.de (I.C.); rainer.schreiber@vkl.uni-regensburg.de (R.S.); khaoula.talbi@vkl.uni-regensburg.de (K.T.); rbenedetto@assay.works (R.B.); 2Department of Pediatrics, Second Faculty of Medicine, University Hospital Motol, Charles University in Prague, Celetná 13, 116 36 Prague, Czech Republic; tereza.dousova@fnmotol.cz; 3Department of Pathology, University of Leuven, UZ Leuven Campus Gasthuisberg, Herestraat 49, B-3000 Leuven, Belgium; erik.verbeken@kuleuven.be (E.K.V.); christiane.deboeck@uzleuven.be (K.D.B.); 4Iaterion, Oakland, CA 94611, USA; isaac@iaterion.com

**Keywords:** TMEM16A, ETX001, Eact, brevenal, asthma, cystic fibrosis, airways

## Abstract

Activation of the Ca^2+^ activated Cl^−^ channel TMEM16A is proposed as a treatment in inflammatory airway disease. It is assumed that activation of TMEM16A will induce electrolyte secretion, and thus reduce airway mucus plugging and improve mucociliary clearance. A benefit of activation of TMEM16A was shown in vitro and in studies in sheep, but others reported an increase in mucus production and airway contraction by activation of TMEM16A. We analyzed expression of TMEM16A in healthy and inflamed human and mouse airways and examined the consequences of activation or inhibition of TMEM16A in asthmatic mice. TMEM16A was found to be upregulated in the lungs of patients with asthma or cystic fibrosis, as well as in the airways of asthmatic mice. Activation or potentiation of TMEM16A by the compounds Eact or brevenal, respectively, induced acute mucus release from airway goblet cells and induced bronchoconstriction in mice in vivo. In contrast, niclosamide, an inhibitor of TMEM16A, blocked mucus production and mucus secretion in vivo and in vitro. Treatment of airway epithelial cells with niclosamide strongly inhibited expression of the essential transcription factor of Th2-dependent inflammation and goblet cell differentiation, SAM pointed domain-containing ETS-like factor (SPDEF). Activation of TMEM16A in people with inflammatory airway diseases is likely to induce mucus secretion along with airway constriction. In contrast, inhibitors of TMEM16A may suppress pulmonary Th2 inflammation, goblet cell metaplasia, mucus production, and bronchoconstriction, partially by inhibiting expression of SPDEF.

## 1. Introduction

Recent work has proposed activation or potentiation of Ca^2+^ activated TMEM16A (anoctamin 1) chloride ion channels as a mutation-agnostic treatment for people with cystic fibrosis (CF) and other inflammatory airway diseases [1]. This strategy is based on early data showing nucleotide stimulation of nasal Cl^−^ transport [2], and numerous subsequent in vitro and ex vivo studies [3,4,5]. It is assumed that activation of TMEM16A will induce airway electrolyte secretion and lead to reduced airway mucus plugging with an improvement in mucociliary clearance. However, purinergic stimulation of Ca^2+^-dependent Cl^−^ secretion by denufosol or direct Ca^2+^-dependent stimulation by lancovutide (Moli1901) failed to show a benefit for lung function in CF patients [6,7]. Recently, the novel TMEM16A potentiator ETX001 was shown to enhance the activity of TMEM16A and increase fluid secretion in vitro, and to improve mucociliary clearance in noninflamed ovine airways in vivo [1]. The related compound ETD002 is currently being evaluated in a phase 1 clinical trial (www.clinicaltrials.gov).

Available data suggest that TMEM16A is hardly expressed in the airways and airway smooth muscle (ASM) of noninflamed lungs, but is upregulated in CF and asthma in humans, piglets, mice, and guinea pigs [8,9,10,11,12]. Nucleotides such as ATP or UTP are released at higher levels during goblet cell metaplasia, causing enhanced airway mucin secretion [13,14]. Aerosolized UTP was shown to increase mucociliary clearance in CF patients, when applied alone or when co-administrated together with amiloride, the inhibitor of epithelial Na^+^ channels [15,16]. However, the lack of beneficial effects of the TMEM16A-activators denufosol and lancovutide on lung function in CF may be due to enhanced mucus secretion.

Danahay et al. reported no effects of the TMEM16A potentiator ETX001 on airway mucus, bronchoconstriction, or pulmonary arterial contraction [17]. This is surprising, as many studies suggest a role of TMEM16A in the upregulation of mucus production as well as mucus release [12,18,19,20,21]. Moreover, upregulation of TMEM16A expression in ASM was shown to strongly correlate with airway hyperresponsiveness and airway contraction [9,22,23,24]. Notably, asthmatic conditions are frequently found in patients with CF [25]. Finally, TMEM16A is reported to contribute to pulmonary artery contraction, leading to pulmonary arterial hypertension [26,27,28,29], a condition that is prevalent in adults with asthma or cystic fibrosis [30,31]. We therefore analyzed the expression and function of TMEM16A in human lungs, in mouse lungs in vivo, and airway epithelial cells in vitro. We found an enhanced expression of TMEM16A in submucosal glands, ASM, and pulmonary blood vessels in CF and asthmatic lungs. The data support a role of TMEM16A in mucus production. Activation or potentiation of TMEM16A induced acute mucus secretion with airway contraction in OVA-challenged mice, while the TMEM16A inhibitor niclosamide antagonized mucus production and secretion. Thus, activation of TMEM16A in inflammatory airway diseases could worsen clinical symptoms.

## 2. Results

### 2.1. Expression of TMEM16A in Healthy, Asthmatic, and CF Lungs

In noninflamed human lungs, TMEM16A was either only sparsely or not at all expressed in both surface epithelium and submucosal glands. In contrast, expression of TMEM16A was detected in submucosal glands of asthma patients and people with CF (Figure 1). CF submucosal glands were often found to be hypertrophic. Expression of TMEM16A is also enhanced in ASM of asthma patients and in people with CF, and appears to be enhanced in the endothelium of pulmonary blood vessels (Appendix A).

### 2.2. Acute Activation of TMEM16A by Eact Induces Mucus Release and Airway Constriction

Eact is an activator of TMEM16A that may also activate other ion channels such as TRPV4 [32,33,34]. We found earlier that Eact potently activates TMEM16A overexpressed in HEK293 cells, with only a slight increase in intracellular Ca^2+^ levels [33]. Eact was applied to ovalbumin (OVA)-sensitized asthmatic mice, which present pronounced goblet cell metaplasia (Figure 2A, left panel). Acute application of Eact by intratracheal instillation induced the release of mucus from goblet cells, which caused accumulation of mucus in airways (Figure 2A, right panel, Figure 2B). Moreover, by analyzing airway cross sections, we detected a pronounced narrowing of the airways due to Eact-induced airway contraction (Figure 2A,C).

### 2.3. Mucus Production and Mucus Secretion Is Strongly Inhibited by Niclosamide

We performed in-depth analysis of mouse lungs using stitching microscopy. The data show a low number of mucus producing cells in control airways (Figure 3, upper panel, Figure 4). In contrast, OVA-sensitized mice demonstrated massive goblet cell metaplasia and accumulation of intraluminal mucus (Figure 3, middle panel, Figure 4). Application of the TMEM16A inhibitor niclosamide for 4 days strongly reduced goblet cell metaplasia, mucus production, and accumulation of mucus in the airways of OVA-sensitized mice (Figure 3, lower panel, Figure 4). The present results suggest beneficial effects of niclosamide, i.e., inhibition of TMEM16A in inflammatory airway diseases, comparable to the results in earlier reports [9,35]. Previous reports also demonstrated relaxation of airways by niclosamide, along with reduced cytokine release [9,35]. It is therefore likely that niclosamide will be beneficial for the treatment of inflammatory airway diseases, including Covid-19 [36]. Additional work will be required to analyze in more detail the effects of niclosamide on the release of other Th1, Th2, and Th17 cytokines. Although niclosamide was shown to dampen intracellular Ca^2+^ signals, we did not find evidence for reduced ciliary stroke amplitude or attenuated mucociliary clearance by niclosamide [33,35] (Appendix A).

### 2.4. Expression of MUC5AC and Activation of TMEM16A in Calu-3 Cells Are Inhibited by Niclosamide

We further analyzed expression of MUC5AC induced by the Th2-cytokine IL-13 in Calu-3 human submucosal epithelial cells (Figure 5). Similar to the findings in mice in vivo, niclosamide potently inhibited MUC5AC expression [37] (Figure 5A,B). Attenuation of mucus production was paralleled by inhibition of TMEM16A whole-cell currents (Figure 5C,D). It should be noted that Calu-3 cells express significant amounts of TMEM16A, particularly after exposure to IL-13 [37]. Treatment with niclosamide not only inhibited TMEM16A currents, but also attenuated TMEM16A expression (Figure 6A). These results obtained in airway cells correspond well to the inhibition of TMEM16A expression by niclosamide and other TMEM16A blockers, such as Ani9 or benzbromarone, in mouse kidneys [38]. Thus, TMEM16A inhibitors not only block Cl^−^ currents, but also inhibit expression of TMEM16A in long-term treatment [38].

### 2.5. Niclosamide Inhibits Expression of TMEM16A, MUC5AC, and SPDEF in Calu-3 Cells

Because long-term treatment with TMEM16A inhibitors suppresses expression of TMEM16A, we asked whether niclosamide affects the transcription of proteins relevant to airway inflammation. In fact, in IL-13-treated Calu-3 cells, transcription of MUC5AC and TMEM16A were clearly blocked by niclosamide (Figure 6B,C). SAM pointed domain-containing ETS transcription factor (SPDEF) is the central integrator of goblet cell differentiation and pulmonary Th2 inflammation [39,40,41]. We also found a pronounced inhibition of SPDEF-mRNA expression by treatment with niclosamide (Figure 6C). Inhibition of SPDEF expression may therefore be a crucial mechanism for the anti-inflammatory effects observed from niclosamide.

### 2.6. Potentiation of TMEM16A Whole-Cell Currents by Brevenal Released Airway Mucus and Caused Bronchoconstriction

Eact has been proposed to be an activator of Ca^2+^-permeable TRPV4 channels [34], which could suggest that the Eact-induced changes observed in the present report, such as bronchoconstriction and mucus release, are caused by a TMEM16A-independent mechanism. Although our previous study did not show a significant increase in the intracellular Ca^2+^ concentration by Eact [33], we nevertheless felt that it was important to rule out this possibility by examining the effects of another putative activator of TMEM16A.

Brevenal is a compound isolated from the marine dinoflagellate Karenia brevis [42]. It was shown to counteract bronchoconstriction induced by brevetoxin in sheep lungs. Brevenal was therefore even proposed as a potential drug to treat mucociliary dysfunctions [42]. In whole-cell patch clamp experiments, we examined the effects of brevenal on TMEM16A whole-cell currents expressed endogenously in CFBE airway epithelial cells. The experiments were performed in the presence of TRAM-34 (25 nM), to block unwanted potential activation of Ca^2+^-activated K^+^ channels. Notably, a low concentration (5–500 nM) of brevenal enhanced basal whole-cell currents. Concentration-dependent activation of TMEM16A Cl^−^ currents by stimulation of purinergic P2Y_2_ receptors with ATP was strongly potentiated by brevenal (Figure 7A–D). Brevenal enhances the basal activity of TMEM16A and facilitates activation of TMEM16A through purinergic stimulation with ATP. The compound therefore may act as an activator and potentiator of TMEM16A. Brevenal appears to enhance the Ca^2+^ sensitivity of TMEM16A, as suggested by enhanced and stable time-dependent current activation at depolarized clamp voltages and enhanced activation of TMEM16A inward currents at negative clamp voltages (Figure 7).

To demonstrate that TMEM16A is responsible for ATP-activated currents and their potentiation by brevenal, TMEM16A expression was knocked down by siRNA (Figure 7, lower recordings, Appendix A). Cells lacking expression of TMEM16A showed a complete absence of ATP-activated whole-cell currents, clearly indicating the exclusive activation of TMEM16A currents by ATP (Figure 7E,F). Moreover, brevenal was no longer able to potentiate activation of TMEM16A Cl^−^ currents. Taken together, these results indicate that brevenal activates basal activity and potentiates purinergic Ca^2+^-dependent activation of TMEM16A.

Brevenal could potentiate TMEM16A by releasing Ca^2+^ from endoplasmic reticulum (ER) Ca^2+^ stores, or could potentiate ATP-induced Ca^2+^ store release. Alternatively, brevenal may enhance the sensitivity of TMEM16A for intracellular Ca^2+^. We examined whether brevenal activates TMEM16A by releasing Ca^2+^ from ER stores, using the cytosolic Ca^2+^ sensor Fura-2. Brevenal neither increased basal intracellular Ca^2+^ concentrations in CFBE cells, nor augmented ATP-induced increase in intracellular Ca^2+^ (Figure 8A–C). We therefore conclude that brevenal potentiates activation of TMEM16A, probably by enhancing its Ca^2+^ sensitivity.

In order to examine how the TMEM16A potentiator brevenal affects airway function, we applied brevenal acutely to mouse airways by tracheal instillation. During the application of brevenal, we observed that some animals developed severe breathing problems. Briefly after the application of brevenal, the mice were sacrificed and their lungs were stained for mucus. Alcian blue staining revealed reduced intracellular mucus in airway goblet cells after treatment with brevenal, suggesting acute release of mucus (Figure 8D,E). Moreover, brevenal-exposed airways appeared to be contracted. Analysis of airway cross sections indicated reduced airway diameters, which suggests airway contraction induced by brevenal (Figure 8D,F). Additional in vivo studies are therefore warranted before TMEM16A activators or potentiators are considered for the treatment of inflammatory airway diseases.

## 3. Discussion

In the present study, we examined the role of the Ca^2+^-activated Cl^−^ channel TMEM16A in mouse airways and human airway epithelial cells. TMEM16A is regarded as a secretory Cl^−^ channel that might compensate for the absence of CFTR-mediated Cl^−^ secretion in the airways of people with cystic fibrosis. While TMEM16A is almost undetectable in normal human airways using a highly specific diagnostic antibody [43], it was clearly detectable in asthmatic lungs and the lungs of CF patients. The surface (or superficial) airway epithelium showed very little expression of TMEM16A in contrast to the apical membrane of submucosal glands. A TMEM16A-based therapeutic strategy should therefore probably target TMEM16A in the submucosal glands, rather than in the surface epithelium. Secretion of both fluid and mucus takes place in submucosal glands [8,44,45]. TMEM16A was detected all along the tubular submucosal glands, suggesting that it may contribute to both CFTR-dependent fluid secretion and mucus secretion. However, because ATP-induced steady fluid secretion is essentially due to CFTR and not by TMEM16A [46], activation of TMEM16A to induce fluid secretion in CF may be ineffective. In contrast, mucus secretion is induced due to upregulation of TMEM16A in goblet cells, which is supports mucus secretion [21]. Thus, TMEM16A-mediated mucus production [37] and mucus release [21] dominate in inflamed airways. Pharmacologic activation of TMEM16A to induce Cl^−^ secretion in CF lungs could therefore be problematic. Additional work in vivo in CF animal models or in biopsies ex vivo is required to quantify the contribution of TMEM16A to mucus and fluid secretion. Exclusive in vitro experiments with cultured airway cells will not provide definitive answers as in vitro monolayers do not maintain the complexity of naïve airways and do not form submucosal glands.

We have shown the expression of TMEM16A in airway smooth muscle in lung sections of patients with asthma or CF. Together with numerous previous studies, there is clear evidence that TMEM16A contributes essentially to airway hyperresponsiveness [9,19,22,24,35]. Rather surprisingly, the TMEM16A potentiator ETX001 apparently did not contract isolated bronchi or arteries of human donors without asthma, CF, or pulmonary hypertension [17]. However, given the fact that expression of TMEM16A in noninflamed human lungs is very low, but upregulated in the lungs of patients with asthma, CF, or pulmonary hypertension, the effects of potentiators or activators should be examined in the lungs of affected patients or in corresponding animal models [21,35,47,48]. Along this line, expression of TMEM16A was upregulated in arterial endothelial cells of CF and asthma lungs, probably due to local hypoxia. TMEM16A overexpression in pulmonary arterial endothelial cells contributes to idiopathic pulmonary hypertension [28,29], and upregulation of TMEM16A in CF lungs is very likely to contribute to PH and decreased survival in severe CF [31].

The present data indicate that application of Eact, the activator of TMEM16A [32], acutely releases airway mucus and contracts airways in vivo. We observed that some of the animals presented with severe breathing problems after the application of Eact. Although we did not observe much of an increase in intracellular Ca^2+^ by Eact in an earlier study [33], Eact may be able to directly increase intracellular Ca^2+^, e.g., by activating transient receptor potential (TRP) channels [34], and may therefore cause pulmonary effects independent of activation of TMEM16A. However, inhibition of TMEM16A by niclosamide completely blocked Eact-induced activation of TMEM16A (Appendix A). We also examined the effects of brevenal, another putative activator of TMEM16A [49]. In the present study, we found that brevenal indeed enhances basal activity of TMEM16A and facilitates activation of TMEM16A through purinergic stimulation with ATP. Moreover, we found no evidence for a direct or indirect rise in intracellular Ca^2+^ by brevenal. Nevertheless, it could still be that brevenal enhances Ca^2+^ very locally and only in close proximity to TMEM16A, because we used Fura-2 to measure global intracellular Ca^2+^ levels. This interesting compound may therefore acts as an activator and potentiator of TMEM16A expressed endogenously in human airway epithelial cells (Figure 7 and Figure 8). Brevenal appears to enhance the Ca^2+^ sensitivity of TMEM16A, which is reflected by enhanced time-dependent current activation and enhanced activation of TMEM16A inward currents [50] (Figure 7). Similarly to Eact, brevenal also caused acute mucus release and bronchoconstriction in vivo, which caused severe breathing problems in several animals. The present results therefore strongly contraindicate the use of brevenal in inflammatory airway diseases, unlike two recent reports which promote brevenal as a treatment in chronic respiratory disease [42,51]. In contrast to these reports, our study was done in animals with induced asthma, which leads to goblet cell metaplasia and upregulation of TMEM16A and a number of proinflammatory proteins, including SPDEF. Moreover, and as shown in Figure 1, noninflamed lung tissue and macrophages [52] express very little or no TMEM16A.

Niclosamide, in contrast, strongly attenuated goblet cell metaplasia in the present and previous reports [35]. The TMEM16A blocker niflumic acid removed airway mucus excess and improved survival in a rat pneumonia model [53]. As TMEM16A is upregulated in ASM and in the arterial endothelium, niclosamide-induced inhibition of TMEM16A is a promising new treatment in CF, IPH, and asthma [27,29,35]. Moreover, niclosamide is currently being investigated as a drug for the treatment of the viral inflammatory lung disease COVID-19 [54]. We have shown that niclosamide inhibits expression of SPDEF, a central integrator in inflammatory lung diseases [39]. Although activation/potentiation of TMEM16A may induce airway Cl^−^ secretion in submucosal glands, parallel increase in mucus production/secretion, ASM contraction and pulmonary arterial blood pressure may compromise lung function in people with inflammatory airway diseases. Taken together, acute activation or potentiation of TMEM16A is likely to induce adverse effects in inflamed airways. Therefore, additional studies in vivo should be performed before these drugs are further evaluated in humans.

## 4. Materials and Methods

Animals and treatments: Allergen challenge of mice has been described in Schreiber et al. [55]. In brief, mice were sensitized to ovalbumin (OVA; Sigma-Aldrich, St. Louis, MO, USA) by intraperitoneal (I.P.) injection of 100 µg OVA in 100 µL aluminium hydroxide gel adjuvant (InvivoGen, San Diego, CA, USA) on days 0 and 14. From days 21 to 23, mice were anesthetized and challenged to OVA by intratracheal (I.T.) instillation of 50 µg OVA in 100 µL saline. Control mice were sham sensitized with the adjuvant aluminum hydroxide gel and challenged to saline by I.T. instillation. TMEM16A activator Eact (4.8 µg/100 µL saline) was administered by I.T. instillation 4 h before animal sacrifice on day 26. TMEM16A inhibitor niclosamide (30 µM in 100 µL saline) was administered by I.T. instillation from days 21 to 25, and animals were sacrificed on day 26. The TMEM16A potentiator brevenal (3.2 µg/100 µL saline) was administered to non-sensitized/challenged control mice by I.T. instillation 10 min before animal sacrifice. Control I.T. instillation was performed with saline alone. All animal experiments complied with the general guidelines for animal research, in accordance with the United Kingdom Animals Act, 1986, and associated guidelines, and EU Directive 2010/63/EU for animal experiments. All animal experiments were approved by the local Ethics Committee of the Government of Unterfranken/Wurzburg/Germany (AZ: 55.2-2532-2-677) and were conducted according to the guidelines of the American Physiologic Society and German Law for the Welfare of Animals.

*Cell culture and treatments:* All cells were grown at 37 °C in a humidified atmosphere with 5% (*v*/*v*) CO_2_. Culture conditions of Calu-3 and CFBE cells have been described earlier [37]. In brief, airway epithelial cells were grown in DMEM/Ham’s F-12 with L-Glutamine medium supplemented with 10% (*v*/*v*) fetal bovine serum (FBS), 1% (*v*/*v*) L-glutamine 200 mM and 1% (*v*/*v*) HEPES 1M (all from Capricorn Scientific, Ebsdorfergrund, Germany). CFBE parental cells were grown in MEM with Earle’s Salts with L-Glutamine medium (Capricorn Scientific, Ebsdorfergrund, Germany) supplemented with 10% FBS. Cells were treated with IL-13 (20 ng/mL; Enzo Life Sciences, Lörrach, Germany) for 72 h in Opti-MEM Reduced Serum Medium (Gibco/Thermo Fisher Scientific, Waltham, MA, USA); treatment was refreshed every day. Niclosamide (ethanolamine salt, 1 µM) was applied simultaneously with IL-13 for MUC5AC staining/ western blot/ PCR experiments or 1 h before measurements for patch clamp experiments. Cells were pre-incubated with brevenal (500 nM) for 15 min for patch clamp experiments or 5 h for Ca^2+^ measurements.

Knockdown of TMEM16A in CFBE parental cells was performed by transfecting siTMEM16A (5-CCUGUACGAAGAGUGGGCACGCUAU-3, Invitrogen, Carlsbad, CA, USA) using standard protocols for Lipofectamine 3000 (Invitrogen, Carlsbad, CA, USA). Scrambled siRNA (Silencer^®^ Select Negative Control siRNA #1, Ambion, Austin, TX, USA) was transfected as negative control. All experiments were performed 72 h after transfection.

*RT-PCR:* For semi-quantitative RT-PCR, total RNA from Calu-3 airway epithelial cells was isolated using NucleoSpin RNA II columns (Macherey-Nagel, Düren, Germany). Reverse transcription of RNA and polymerase chain reaction (PCR) were described in an earlier report [37]. Total RNA (0.5 µg / 25 µL reaction) was reverse-transcribed using random primer (Promega, Mannheim, Germany) and M-MLV Reverse Transcriptase RNase H Minus (Promega, Mannheim, Germany). Each RT-PCR reaction contained sense and antisense primers (0.5 µM) (Table 1), 0.5 µL cDNA, and GoTaq Polymerase (Promega, Mannheim, Germany). After 2 min at 95 °C, cDNA was amplified (20–30 cycles) for 30 s at 95 °C, 30 s at 57 °C, and 1 min at 72 °C. PCR products were visualized by loading on peqGREEN-(Peqlab, Düsseldorf, Germany) containing agarose gels and analyzed using ImageJ.

*Western blotting:* Protein was isolated from cells using a lysis buffer containing 25 mM Tris-HCl pH 7.4, 150 mM NaCl, 1 mM EDTA, 5% glycerol, 0.43% Nonidet P-40, 100 mM dithiothreitol (both from PanReac AppliChem, Barcelona, Spain), and 1× protease inhibitor mixture (Roche, Basel, Switzerland). Protein separation and blotting were performed as described in an earlier report [37]. Proteins were separated by 8.5% SDS-PAGE and transferred to a PVDF membrane (GE Healthcare, Munich, Germany). Membranes were incubated with primary rabbit monoclonal [SP31] anti-TMEM16A antibody (#ab64085; Abcam, Cambridge, UK; 1:500 in 1% (*w*/*v*) NFM/TBS-T) overnight at 4 °C. A rabbit polyclonal anti-β-actin antibody (#A2066; Sigma-Aldrich, St. Louis, MO, USA; 1:10 000 in 5% (*w*/*v*) NFM/TBS-T) was used for loading control. Afterwards, membranes were incubated with horseradish peroxidase (HRP)-conjugated goat polyclonal anti-rabbit secondary antibody (#31460; Invitrogen, Carlsbad, CA, USA) at room temperature for 2 h, and immunoreactive signals were visualized using a SuperSignal HRP Chemiluminescence Substrate detection kit (#34577; Thermo Fisher Scientific, Waltham, MA, USA).

*Immunohistochemistry, immunocytochemistry, airway cross sections, and stitching microscopy:* Staining and analysis of human lung tissue slides were approved by the ethics committee of the University Hospital Motol, Charles University in Prague, Prague, Czech Republic (approval number EK-853//18, date of approval 18 July 2018). Paraffin embedded human lung sections (5 µm) were deparaffinized with xylene and rehydrated through a series of ethanol. Antigen retrieval was performed in pre-heated Tris-EDTA buffer (pH 9.0) for 15 min, using a microwave. After cooling down to room temperature and appropriate washing, sections were blocked with 5% IgG-free bovine serum albumin (BSA) and 0.05% Triton X-100 in PBS with 0.05% Tween-20 (PBS-T) for 1 h at room temperature. Primary antibodies were diluted in 1% IgG-free BSA in PBS-T and incubated overnight at 4 °C. Rabbit polyclonal anti-human TMEM16A antibody (1:100; raised against YLKLKQQSPPDHEECVKRKQR, aa 688-708; Davids Biotechnology, Regensburg, Germany) was used for immunofluorescence staining. After washing steps, sections were incubated in Alexa Fluor 488-labeled donkey anti-rabbit IgG (1:400; Invivogen Europe, Toulouse, France) and counterstained with Hoe33342 (1 mM, 1:200; AppliChem, Darmstadt, Germany) for 1 h at room temperature. After washing, coverslips were then mounted in fluorescence mounting medium (Dako, Hamburg, Germany). For 3,3′-diaminobenzidine (DAB) staining, quenching of endogenous peroxidase was performed before blocking step with 1% hydrogen peroxide in ethanol for 15 min. The primary antibody, anti-DOG1/TMEM16A (SP31, Novus Biologicals, USA) was incubated overnight at 4 °C. After washing, biotinylated donkey anti-rabbit IgG (Santa Cruz, Heidelberg, Germany) at 1:500 dilution was applied for 1 h at room temperature. Sections were washed in PBS-T and were then incubated with avidin-peroxidase complex (Vectastain kit, Vector laboratories) for 1 h at room temperature. The peroxidase was then developed by DAB (Sigma, Taufkirchen, Germany). Slides were counterstained with hematoxylin. Immunofluorescence and immunohistochemistry were detected with an Axio Observer microscope equipped with Axiocams 503 mono and 305 color, ApoTome.2, and ZEN 3.0 (blue edition) software (Zeiss, Oberkochen, Germany). Cross sectional areas were determined by automatic marking and analysis of airway lumens using Zen software and Axio Observer microscope equipped with AxioCam and ApoTome2. Stitching microscopy was performed using motorized Axio Observer and Zen software.

For MUC5AC stainings, Calu-3 airway cells seeded onto glass coverslips were fixed with 4% PFA/PBS for 10 min at room temperature, washed in PBS with Ca^2+^ and Mg^2+^ (PBS^++^), incubated in 0.5% Triton X-100/PBS^++^ for 10 min at room temperature, washed, blocked with 1% BSA/PBS^++^ for 40 min at room temperature, incubated with mouse monoclonal anti-MUC5AC antibody (1:300 in 1%BSA/PBS^++^; #ab3649; Abcam, Cambridge, UK) for 1 h at 37 °C, washed, incubated with Alexa Fluor 488-labeled donkey anti-mouse IgG (1:300 in 1% BSA/PBS^++^; Invitrogen, Carlsbad, CA, USA) and counterstained with Hoe33342 (1:200) for 1 h at room temperature. Cells were then washed and mounted in fluorescence mounting medium. MUC5AC staining was quantified using ImageJ. Additional details are provided in an online data supplement.

*Mucus staining and analysis of mucus release and airway contraction:* Mouse airways were fixed by transcardial perfusion and lung perfusion by tracheal instillation via tracheostomy of fixative solution containing 4% PFA in PBS. Tissues were left in fixative solution overnight and embedded in paraffin the next day. 5 µm cuts were deparaffinized, stained with standard Alcian blue solution, and counterstained with Nuclear Fast Red solution (Sigma-Aldrich, St. Louis, MO, USA). After the dehydration and clearing steps, whole mouse lungs or sections were mounted in DePeX mounting medium (SERVA Electrophoresis, Heidelberg, Germany). Stainings were assessed by light microscopy. Stitching microscopy was used to analyze whole mouse lungs. Mucus-stained and cross sectional areas were determined using ImageJ.

*Patch Clamp:* Calu-3 and CFBE cells were patch clamped after growing on coated glass coverslips for 2–3 days. Whole-cell patch clamp techniques and data analysis have been described earlier [46]. In brief, patch pipettes were filled with a cytosolic-like solution containing (in mM): KCl 30, K-Gluconate 95, NaH_2_PO_4_ 1.2, Na_2_HPO_4_ 4.8, EGTA 1, Ca-Gluconate 0.758, MgCl_2_ 1.03, D-Glucose 5, ATP 3; pH 7.2. The intracellular (pipette) Ca^2+^ activity was 0.1 µM. Fast whole-cell current recordings were performed as described recently [56]. The bath was perfused continuously with standard bicarbonate-free Ringer’s solution (in mM: NaCl 145, KH_2_PO_4_ 0.4, K_2_HPO_4_ • 3 H_2_0 1.6, Glucose 5, MgCl_2_ • 6 H_2_0 1, Ca-Gluconate • 1 H_2_0 1.3) at a rate of 8 mL/min. Patch pipettes had an input resistance of 2–4 MΩ and whole-cell currents were corrected for serial resistance. Currents were recorded using a patch clamp amplifier (EPC 7; List Medical Electronics, Darmstadt, Germany), the LIH1600 interface, and PULSE software (HEKA, Lambrecht, Germany), as well as Chart software (AD Instruments, Spechbach, Germany). Cells were stimulated with 1, 10 or 100 µM ATP in standard bicarbonate-free Ringer’s solution. Cells were current clamped for most of the time. In regular intervals, membrane voltage (*V*c) was clamped in steps of 20 mV from −100 to +100 mV. The inhibitor of Ca^2+^-activated KCNN4 K^+^ channels, TRAM-34 (100 nM), was present in all patch clamp experiments to avoid potential activation of Ca^2+^-activated K^+^ channels.

*Measurement of [Ca^2+^]_i_^−^:* Cells were seeded on coated glass coverslips and loaded with 2 µM Fura-2, AM Ester (Biotium, Hayward, CA, USA) and 0.02% Pluronic F-127 (Invitrogen, Carlsbad, CA, USA) in standard bicarbonate-free Ringer’s solution for 1 h at room temperature. Measurement of intracellular Ca^2+^ concentrations has been described earlier [46]. Cells were then mounted in a thermostatically controlled imaging chamber adapted to an inverted microscope (Axiovert S100, Zeiss, Oberkochen, Germany), maintained at 37 °C and perfused at a rate of 5 mL/min. Fura-2 was excited at 340/380 nm using a high-speed polychromatic illumination system for microscopic fluorescence measurements (Visitron Systems, Puchheim, Germany), and emission was recorded between 470 and 550 nm using a CoolSnap HQ CCD camera (Roper Scientific, Planegg, Germany/Visitron Systems, Puchheim, Germany). Cells were stimulated with 1, 10 and 100 µM ATP in standard bicarbonate-free Ringer’s solution. Intracellular calcium ([Ca^2+^]_i_) was calculated from the 340/380 nm fluorescence ratio after background subtraction using the formula *[Ca^2+^]_i_ = Kd × (R − R_min_)/(R_max_ − R) × (S_f2_/S_b2_)*, where *R* is the observed fluorescence ratio. The values *R_max_* and *R_min_* (maximum and minimum ratios) and the constant *S_f2_/S_b2_* (ratio between the fluorescence of free and Ca^2+^-bound Fura-2 at 380 nm) were determined using 2 µM ionomycin (Calbiochem, San Diego, CA, USA), 5 µM nigericin (Sigma-Aldrich, St. Louis, MO, USA), 10 µM monensin (Sigma-Aldrich, St. Louis, MO, USA) and 5 mM EGTA (Carl Roth, Karlsruhe, Germany) to equilibrate intracellular and extracellular Ca^2+^ in intact Fura-2-loaded cells. The dissociation constant (*Kd*) for the Fura-2•Ca^2+^ complex was taken as 224 nM [57] (Grynkiewicz G et al., 1985; *A New Generation of Ca^2+^ Indicators with Greatly Improved Fluorescence Properties*). Control of experiment, imaging acquisition, and data analysis were done with the software package MetaFluor (Universal Imaging, Bedford Hills, NY, USA).

*Mucociliary clearance analysis:* Freshly excised tracheas from non-sensitized mice were cut open and mounted onto glass coverslips in a humidified chamber. 1 µm fluorescent beads (FluoSpheres™ #F8823; Invitrogen, Carlsbad, CA, USA) were added to the apical side of the isolated tracheas in standard bicarbonate-free Ringer’s solution. Niclosamide (1 µM) was applied acutely to the tracheal surface in Ringer’s solution. Video imaging of the tracheal surface (Axio Observer, Zeiss, Oberkochen, Germany) allowed for the tracking of fluorescent particle trajectories and speed (analysis by Image J), and assessment of ciliary beating properties such as stroke amplitude and frequency (analysis by ZEN 3.0 (blue edition) software, Zeiss, Oberkochen, Germany). As experimental control, ciliary movement and thus particle clearance were inhibited by cooling tracheas down to 4 °C.

*Materials and statistical analysis:* All compounds used were of the highest available grade of purity and were bought from Sigma-Aldrich (St. Louis, MO, USA), unless indicated otherwise. Brevenal was kindly provided by Dr. Daniel Baden, PhD (UNC, Wilmington, North Carolina, USA). Data are shown as individual traces/representative images and/or as summaries with mean values ± SEM, with the respective number of experiments given in each figure’s legend. For statistical analysis, paired or unpaired Student’s *t*-test or ANOVA were used where appropriate. A *p*-value of <0.05 was accepted as a statistically significant difference.

## Figures and Tables

**Figure 1 ijms-22-07852-f001:**
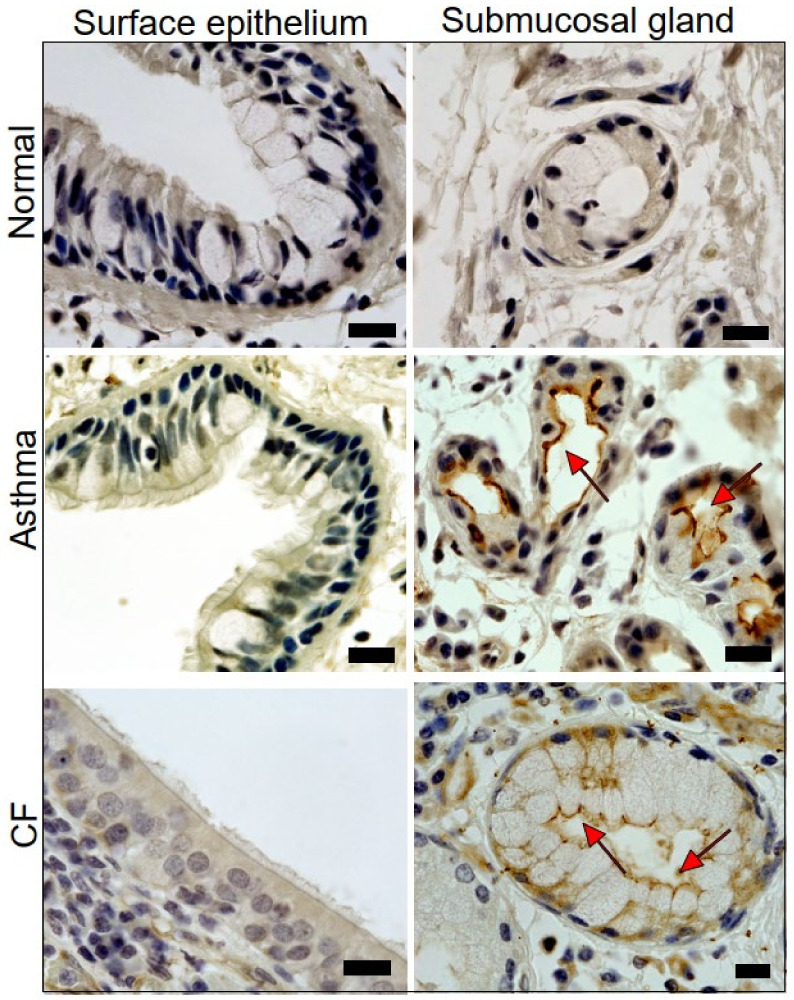
Expression of TMEM16A in human healthy, asthmatic, and CF lungs. Lung slices of normal airways indicate little expression of TMEM16A in both surface airway epithelium and airway submucosal glands. Pronounced expression of TMEM16A in apical membranes of surface epithelial cells and airway submucosal glands of patients with asthma and CF (brown precipitation, DAP staining). Submucosal glands were often found to be hypertrophic. Representative stainings of four patients each. Bars indicate 20 µm.

**Figure 2 ijms-22-07852-f002:**
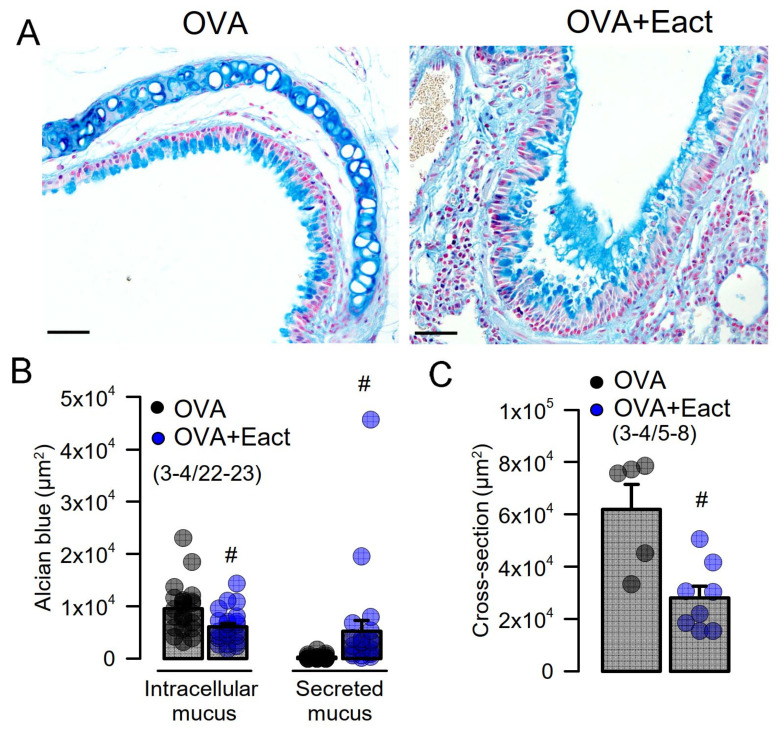
Activation of TMEM16A by Eact induces mucus release and airway contraction. (**A**) Mucus staining by alcian blue in OVA-treated asthmatic mice. Acute application of the activator of TMEM16A, Eact (4.8 µg/mL intratracheal), induced acute release of mucus from goblet cells and increased intraluminal mucus. Bars indicate 50 µm. (**B**) Summary of intracellular and intraluminal mucus. (**C**) Summary of cross sectional area indicating airway contraction by Eact. Mean ± SEM (number of animals/number of measurements). # significant difference when compared to control (*p* < 0.05, unpaired *t*-test).

**Figure 3 ijms-22-07852-f003:**
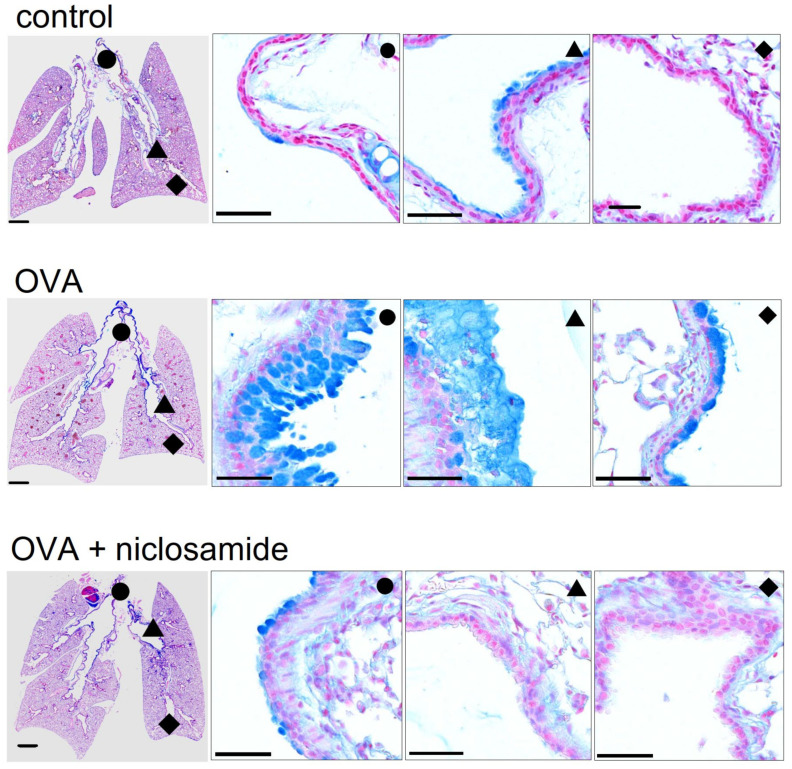
Mucus production and mucus secretion is inhibited by niclosamide. Whole mouse lung analysis by stitching microscopy. Low level mucus production and mucus secretion in proximal (●), central (▲), and peripheral (♦) airways of control mice. Bars = 1000 (very left panels) and 50 µm (higher resolution panels), respectively. Pronounced upregulation of mucus production and intraluminal mucus accumulation in lungs of OVA-treated (asthmatic) mice. Mucus production and mucus secretion is strongly suppressed in asthmatic mice treated with niclosamide (13 µg/mL, 4 days).

**Figure 4 ijms-22-07852-f004:**
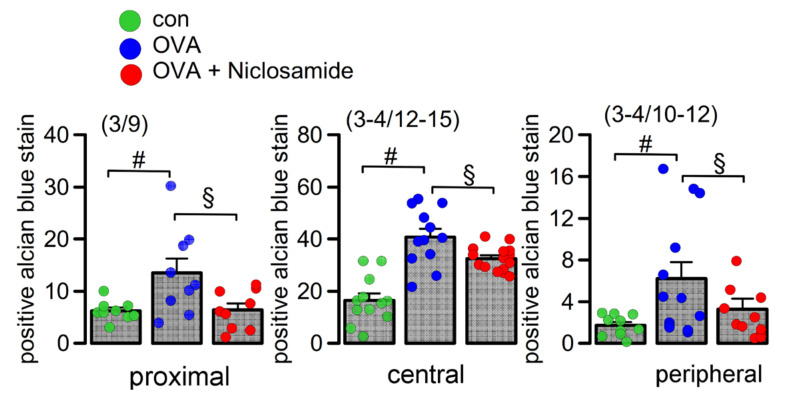
Quantification of airway mucus. Quantification of mucus in proximal, central, and peripheral airways. Mean ± SEM (Number of mice/number of airways analyzed). ^#^ significant difference when compared to control (*p* < 0.05, ANOVA). ^§^ significant difference when compared to OVA (*p* < 0.05, ANOVA).

**Figure 5 ijms-22-07852-f005:**
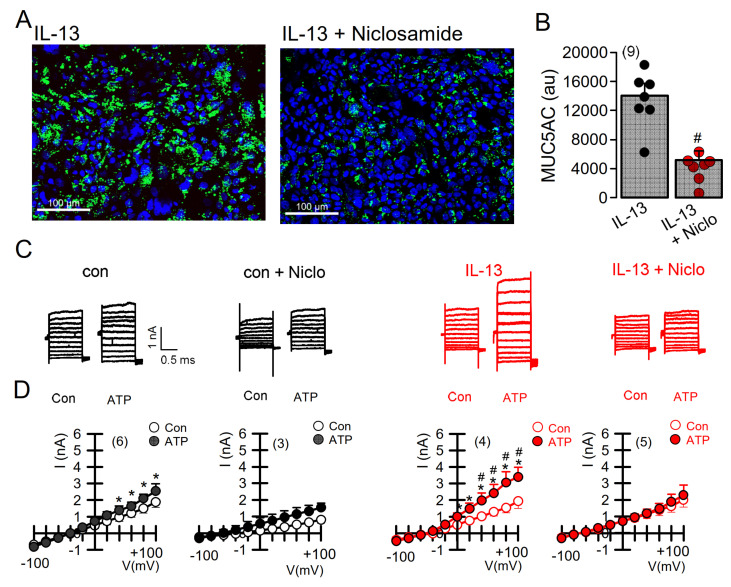
Expression of MUC5AC and activation of TMEM16A in Calu-3 airway epithelial cells is inhibited by niclosamide. (**A**) Expression of MUC5AC induced by IL-13 (20 ng/mL; 72 h) in Calu-3 airway epithelial cells was inhibited by simultaneous incubation with niclosamide (1 µM). Bar = 100 µm. (**B**) Quantification of MUC5AC expression indicating inhibition by niclosamide (Niclo). (**C**) Current overlays from whole-cell patch clamp experiments before and after induction of MUC5AC expression by IL-13. Activation of whole-cell currents by purinergic stimulation (ATP, 100 µM) was enhanced by IL-13, which was completely inhibited by acute application of niclosamide (1 µM). (**D**) Corresponding current/voltage relationships. The inhibitor of Ca^2+^-activated KCNN4 K^+^ channels, TRAM-34 (100 nM), was present in all patch clamp experiments to avoid potential activation of Ca^2+^-activated K^+^ channels. Mean ± SEM (number of cells). * significant activation by ATP (*p* < 0.05, paired *t*-test). # significant difference when compared to the absence of IL-13 (*p* < 0.05, unpaired *t*-test).

**Figure 6 ijms-22-07852-f006:**
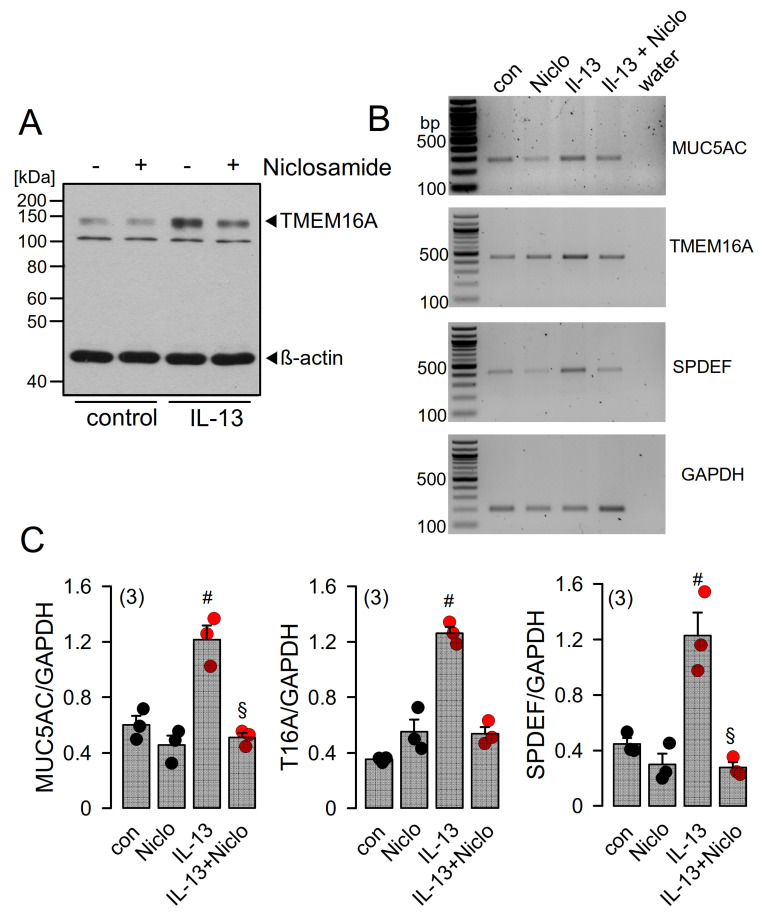
Niclosamide inhibits expression of MUC5AC and SPDEF in Calu-3 cells. (**A**) Western blot indicating upregulation of TMEM16A in Calu-3 cells by IL-13 (20 ng/mL, 72 h) and inhibition of expression by niclosamide (1 µM). Blots were performed in triplicates. (**B**) RT-PCR analysis of the expression of MUC5AC, TMEM16A, and SAM pointed domain-containing ETS transcription factor (SPDEF) in Calu-3 airway epithelial cells. SPDEF is an integrator of goblet cell differentiation and pulmonary Th2 inflammation. (**C**) Low cycle numbers (20×) were chosen for quantification of expression by relating specific signals to expression of the housekeeper protein GAPDH. IL-13 (20 ng/mL) leads to upregulation of expression of MUC5AC, TMEM16A, and SPDEF. Niclosamide (Niclo, 1 µM, 72 h) strongly inhibits expression of MUC5AC, TMEM16A, and SPDEF. Mean ± SEM (number of assays). ^#,§^ significant increase by IL-13 and inhibition by niclosamide, respectively (*p* < 0.05, ANOVA).

**Figure 7 ijms-22-07852-f007:**
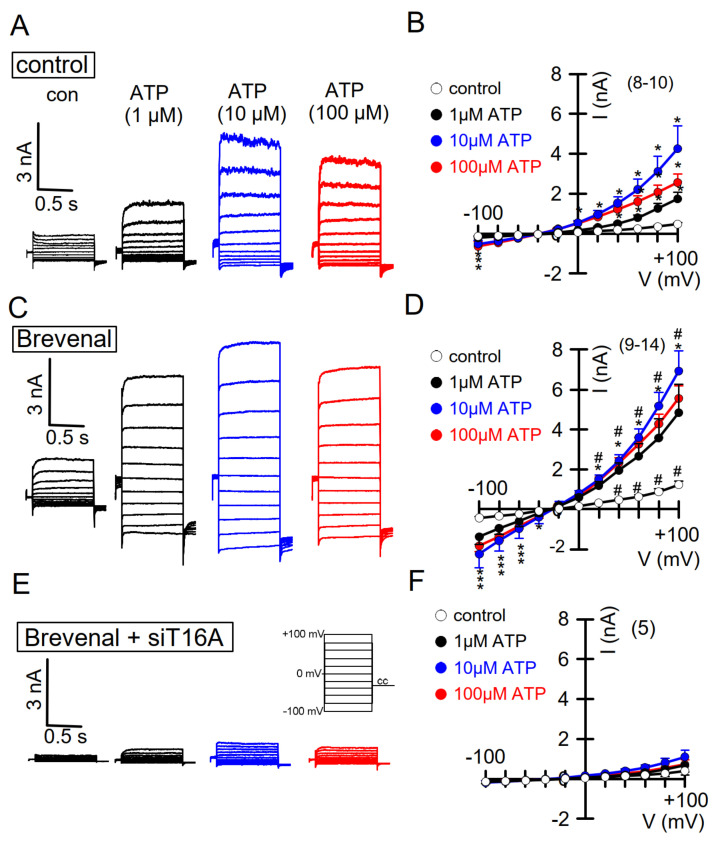
Activation of TMEM16A whole-cell currents is potentiated by brevenal. (**A**) Dose-dependent activation of endogenous TMEM16A whole-cell currents in CFBE airway epithelial cells, by the purinergic agonist ATP. Clamp voltages ± 100 mV in steps of 20 mV. (**B**) Corresponding current/voltage relationships. (**C**,**D**) Activation of whole-cell currents obtained from cells pre-incubated with brevenal (500 nM, 15 min) and corresponding current/voltage relationships. (**E**) Activation of whole-cell currents in brevenal-incubated cells, in which expression of TMEM16A has been knocked down by treatment with siRNA-TMEM16A (c.f. Appendix A). (**F**) Corresponding current/voltage relationships. Mean ± SEM (number of cells). * significant activation by ATP (*p* < 0.05, ANOVA). The inhibitor of Ca^2+^-activated KCNN4 K^+^ channels, TRAM-34 (100 nM), was present in all patch clamp experiments to avoid potential activation of Ca^2+^-activated K^+^ channels. # significant difference when compared to the absence of brevenal (*p* < 0.05, ANOVA). No currents were activated in siRNA-TMEM16A-treated cells.

**Figure 8 ijms-22-07852-f008:**
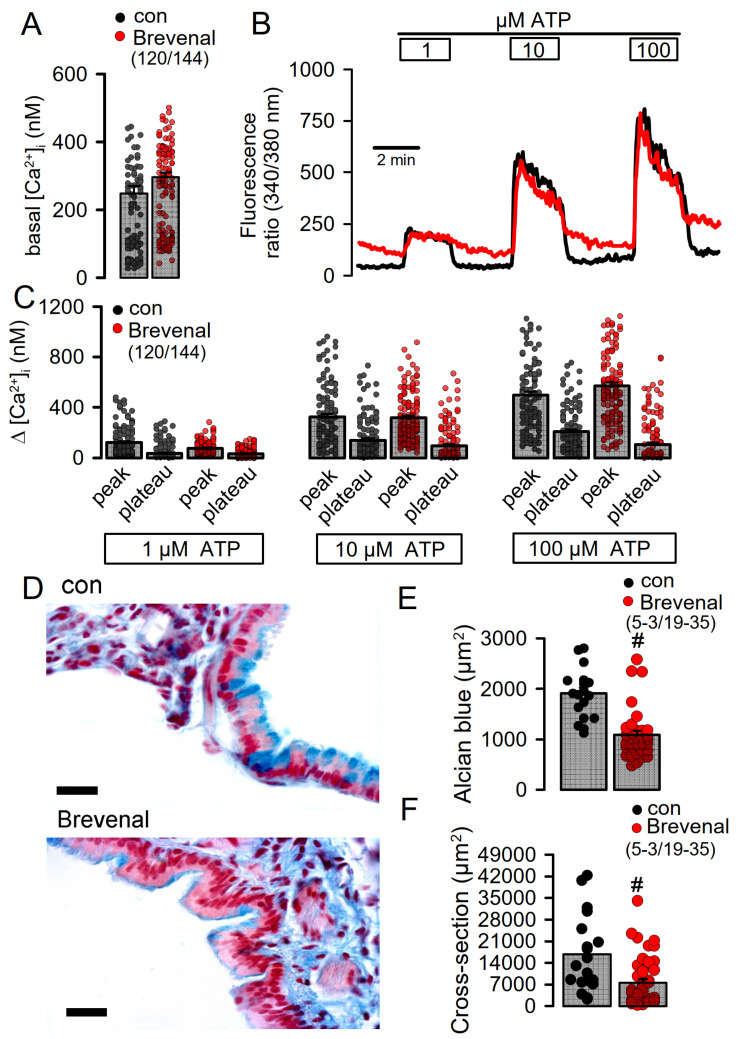
Brevenal does not increase intracellular Ca^2+^ but releases mucus from goblet cells and induces airway contraction. (**A**) Comparable basal intracellular Ca^2+^ levels in airway epithelial cells in the absence or presence of brevenal (500 nM, 15 min preincubation), as measured by the Ca^2+^ sensor Fura-2. (**B**) Concentration-dependent rise in intracellular Ca^2+^ by ATP in the absence or presence of brevenal. (**C**) Concentration-dependent increase in peak and plateau Ca^2+^ by ATP in the absence or presence of brevenal. (**D**) Effects of acute application of brevenal (3.2 µg/100 µL, 10 min) in mouse airways by tracheal instillation. Alcian blue staining of mucus indicated reduced intracellular mucus in airways treated with brevenal. Brevenal-exposed airways appeared contracted. Bars = 15 µm. (**E**) Summary of alcian blue staining in airway epithelial cells reflecting the amount of intracellular mucus, which is significantly reduced upon exposure to brevenal. (**F**) Summary of airway cross sectional area obtained in airways of control mice and mice treated with brevenal, suggesting airway contraction by brevenal. Mean ± SEM; (number of animals/number of measurements). ^#^ significant difference when compared to control mice (*p* < 0.05, ANOVA).

**Table 1 ijms-22-07852-t001:** RT-PCR Primer.

TMEM16A	forward: 5′-CGACTACGTGTACATTTTCCG reverse: 5′-GATTCCGATGTCTTTGGCTC	445 bp
MUC5AC	forward: 5′-GCTCAGCTGTTCTCTGGACG reverse: 5′-GTCACATTCCTCAGCGAGGTC	279 bp
SPDF	forward: 5′-GTGCTCAAGGACATCGAGAC reverse: 5′-CCTAATGAAGCGGCCATAGC	423 bp
GAPDH	forward: 5′-GTATTGGGCGCCTGGTCAC reverse: 5′-CTCCTGGAAGATGGTGATGG	200 bp

## Data Availability

MDPI Research Data Policies at https://www.mdpi.com/ethics.

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
