# Peer review of "Mucus Release and Airway Constriction by TMEM16A May Worsen Pathology in Inflammatory Lung Disease"

_ijms, 2021, doi:10.3390/ijms22157852_

Round 1

Reviewer 1 Report

This manuscript studied the role of the calcium-activated chloride channel TMEM16A in mucus release, airway constriction in various model (mouse and human). One of the objectives of the study was also to verify some of the conclusions of the Danahay’s paper (ref 17). Overall many points need to be addressed, clarified and improved.

Major:

  • The reviewer is not convinced by the results of cross-section measures. Danahay and colleagues used the wire myography method to study muscle contraction ex vivo. This method is well appropriate allowing precise measurement of basal tone, relaxation and constriction. Some results using this method should be added to better studied the effect of pharmacological TMEM16A agonists/antagonists.
  • Although in both CF and asthma lungs, TMEM16A appears expressed above the level observed in uninflamed tissue as show figure 1, authors conclusions should be moderate when analyzing experiments with the OVA-mouse model (a well-accepted model mimicking asthma condition) extrapolating too quickly to the CF condition. The title should be amended too since the present work is not directly related to CF except with fig 1 that only show T16A expression.
  • The second part of the study with brevenal is puzzling for the reviewer. What is the justification of adding these results after describing the effects of niclosamide, IL13 in the OVA model? How both parts are related?
    • Ref 42 showed that brevenal inhibits bronchoconstriction in a sheep model. In their study, authors show the opposite but data are not fully convincing.
    • In a recent work, Keeler et al (2019) – not cited in this work- concluded that brevenal could be an antiflammatory drug (using macrophages and lung epithelial cells). Thus, the effects on lung functions of brevenal are multiple and its mechanism of action not yet clarified. This is questioning any selective effect of brevenal on T16A?
    • Could you show currents with Eact +/- niclo and brevenal+/- niclo
    • The results exploring TMEM16A currents also need some clarifications. For example in Figure 7E, the scale is not adjusted to the amplitude of the recorded currents. Authors concluded in the condition “brevenal in siT16A” that no currents were activated. A quick glance shows this is not the case when comparing the control to ATP.
    • Again in panels A and C, control WCR traces (with and without brevenal) clearly show differences between control and brevenal. Brevenal may activates the current in the absence of ATP. Moreover the current dynamic after ATP 10 and 100µM is different with and without brevenal (in A and C). Could you comment.
    • Are you sure the holding voltage (line 417) is -100mV? What is the equilibrium potential for Cl in your experimental conditions? Please show the protocol used to record your currents.
    • The inward current is clearly more affected than the outward current after brevenal. Could you compare both more precisely ? For example the current amplitude at the assumed potential of -100 mV (fig 7 panel C) is above 3nA but the I/V curves in D do not show that. These experiments need clarifications.
    • The results with Brevenal are not discussed in the discussion. Why?

Minor :

M&M: the method describing how the cross-sectional area is determined is missing

M&M: line 408: no inside-out patch clamp experiments have been done here. It is indicated current-density but all the data are presented only as current (nA). Please carefully check the M&M.

Line 13 : share first authorship? no indication in the author list

Line 62: found in patients

Line 95-97: I guess ref to Fig 2C should be indicated here. The sentence of conclusion is not relevant in this part and too premature since these data are from OVA-animals

Line 100: the activator: repetition in text

Line 102: Bar indicates or Bar indicate

Line 125: Bars…respectively. 2 values but 3 conditions. Please precise

Line 153: inhibited acute…amend here

Line 163: GAPDH

Line 164: SPDEF should be defined above line 161 for example

Line 173: Figure 7? Should be Figure 6 I think

Line 176: Figure 6: do you mean 6B and C?

Line 192-194: add at least one ref for Brevenal here

Line 197-199: why adding TRAM-25? No reference here?  traces without TRAM-25.

Legend Fig E1 and E2: please indicate what mean yellow and red arrows

Legend Fig E3: tracheas

Lines 225-229: sentence should be in the discussion rather than here

Author Response

Reviewer 1

General comment

This manuscript studied the role of the calcium-activated chloride channel TMEM16A in mucus release, airway constriction in various model (mouse and human). One of the objectives of the study was also to verify some of the conclusions of the Danahay’s paper (ref 17). Overall many points need to be addressed, clarified and improved.

Response

We are grateful to this reviewer for putting so much effort into reviewing our manuscript. At numerous occasions we changed the manuscript. The research design and methods are now better justified, the results are more clearly described and the conclusions regarding cystic fibrosis toned down.

Major:

  • The reviewer is not convinced by the results of cross-section measures. Danahay and colleagues used the wire myography method to study muscle contraction ex vivo. This method is well appropriate allowing precise measurement of basal tone, relaxation and constriction. Some results using this method should be added to better studied the effect of pharmacological TMEM16A agonists/antagonists.

Response

We thank this reviewer for his/her valuable suggestion. We agree that wire myography allows for a better quantification of pharmacological effects on airway contraction. This has been already done extensively in our previous study “Miner et al: Drug Repurposing: The Anthelmintics Niclosamide and Nitazoxanide Are Potent TMEM16A Antagonists That Fully Bronchodilate Airways” {Miner, 2019 #7527}. On 85 pages Kent Miner and colleagues present numerous data obtained in wire myograph recordings.

In our previous reports which are all published in good quality journals, we successfully used the present method of cross-sectional analysis (Centeio et al, CLCA1 Regulates Airway Mucus Production and Ion Secretion Through TMEM16A, IJMS, 2021, Cabrita et al, TMEM16A Mediated Mucus Production in Human Airway Epithelial Cells, Am J Respir Cell Mol Biol 2020, Niclosamide repurposed for the treatment of inflammatory airway disease, JCI insight 2019, Benedetto et al, TMEM16A is indispensable for basal mucus secretion in airways and intestine, FASEB J, 2019 {Centeio, 2021 #9084}{Cabrita, 2020 #8761}{Cabrita, 2019 #8317}{Benedetto, 2019 #8088}). The intension of this present study was to demonstrate the effects of activators and inhibitors of TMEM16A in vivo. There are not so many techniques available to measure airway constriction in vivo. We performed additional measurements using a specialized mouse-plethysmograph and determined the so-called enhanced pause (Pen) as in our earlier report {Schreiber, 2008 #3863}. The results from these measurements are in line with the histological cross-sectional analysis. However, the data obtained by this technique are no longer universally accepted {Bates, 2004 #7819}, the were removed from the present manuscript. This reviewer may also note that the purpose of the present study was primarily to analyze the effects of the TMEM16A-inhibitor niclosamide on asthmatic airways in vivo.

  • Although in both CF and asthma lungs, TMEM16A appears expressed above the level observed in uninflamed tissue as show figure 1, authors conclusions should be moderate when analyzing experiments with the OVA-mouse model (a well-accepted model mimicking asthma condition) extrapolating too quickly to the CF condition. The title should be amended too since the present work is not directly related to CF except with fig 1 that only show T16A expression.

Response

We fully agree with the comments of this reviewer and significantly toned down our statements extrapolating the findings in asthmatic mice to CF patients. Moreover, the title has been changed to “Mucus release and airway constriction by TMEM16A may worsen pathology in inflammatory lung disease“. The abstract now only refers to “inflammatory airway disease” but not to cystic fibrosis.

  • The second part of the study with brevenal is puzzling for the reviewer. What is the justification of adding these results after describing the effects of niclosamide, IL13 in the OVA model? How both parts are related?

Response

We apologize for being unclear. We changed the structure of the manuscript and clearly explain why we used the compound brevenal in addition to Eact. Also, the sequence of the figures was changed accordingly: “Eact has been proposed to be an activator of Ca2+ -permeable TRPV4 channels {Genovese, 2019 #8458}, which may suggest that the Eact-induced changes observed in the present report such a bronchoconstriction and mucus release are caused by a TMEM16A-independent mechanism. Although our previous study did not show a significant increase in the intracellular Ca2+ concentration by {Centeio, 2020 #8572}, we nevertheless felt that it is important to rule out this possibility by examining the effects of another putative activator of TMEM16A”.

    • Ref 42 showed that brevenal inhibits bronchoconstriction in a sheep model. In their study, authors show the opposite but data are not fully convincing.

Response

Thank you for bringing this to our attention. The analysis of bronchoconstriction and mucus release induced by brevenal was analyzed as reported in our previous studies {Miner, 2019 #7527}{Centeio, 2021 #9084}{Cabrita, 2020 #8761}{Benedetto, 2019 #8088}{Cabrita, 2019 #8317}. As outlined above, we tried additional measurements using a mouse-plethysmograph to determine the so-called enhanced pause (Pen), but this technique is no longer accepted {Bates, 2004 #7819}. This reviewer may have seen that in the study by Abraham et al, {Abraham, 2005 #8533} the team used brevetoxin to induce airway constriction in sheeps, which was antagonized by brevenal. However, in our study we used mice with a pronounced goblet cell metaplasia, indicative for an airway inflammation. Under these conditions a number of proteins such as TMEM16A, MUC5AC, SPDEF, inflammatory cytokines and many more are upregulated, which was most likely not the case in the study by Abraham et al. As demonstrated in the present report, under conditions of goblet cell metaplasia, brevenal induced a mucus release and caused bronchoconstriction. In fact, several animals developed severe breathing problems during the application of brevenal. Brevenal-induced mucus release and bronchoconstriction correspond well to the fact that brevenal augmented basal and ATP-induced, i.e. Ca2+ mediated activation of TMEM16A. The amount of available purified brevenal unfortunately was not enough to continue with a larger series of in vitro experiments. A discussion on this has now been included into the manuscript.

    • In a recent work, Keeler et al (2019) – not cited in this work- concluded that brevenal could be an antiflammatory drug (using macrophages and lung epithelial cells). Thus, the effects on lung functions of brevenal are multiple and its mechanism of action not yet clarified. This is questioning any selective effect of brevenal on T16A?

Response

We fully agree with this reviewer that the paper by Keeler et al suggests that brevenal may induce multiple effects depending on the cell type, and is not specific for TMEM16A. However, the different results are very likely due to the fact that in their study the cells or tissues did not express TMEM16A. This is now discussed and the paper by Keeler et al is cited.

    • Could you show currents with Eact +/- niclo and brevenal+/- niclo

Response

We have now included whole cell patch clamp experiments with TMEM16A overexpressed in HEK293 cells that demonstrate inhibition of Eact-activated TMEM16A whole cell currents by niclosamide (Fig. E5). Unfortunately we have no longer brevenal available to do the same type of experiment with this type of TMEM16A-activator. However, because we demonstrated earlier that niclosamide is a potent inhibitor of TMEM16A {Miner, 2019 #7527}{Centeio, 2020 #8572}{Cabrita, 2019 #8317} and because activation of TMEM16A by brevenal is completely inhibited by siRNA-knockdown of TMEM16A (Fig. 7E), it can be predicated that niclosamide also inhibits brevenal-activation of TMEM16A in HEK293 cells.

    • The results exploring TMEM16A currents also need some clarifications. For example in Figure 7E, the scale is not adjusted to the amplitude of the recorded currents. Authors concluded in the condition “brevenal in siT16A” that no currents were activated. A quick glance shows this is not the case when comparing the control to ATP.

Response

We apologize for being unclear: siRNA-knockdown of endogenous TMEM16A in CFBE cells inhibited basal whole cell currents as well as ATP-activated whole cell currents very prominently. The scale is correct and applies to Fig. 7A, 7C, and 7E. To make this clear, we added the scale also to the other panels. No significant current activation was detected in cells treated with si-TMEM16A.

    • Again in panels A and C, control WCR traces (with and without brevenal) clearly show differences between control and brevenal. Brevenal may activates the current in the absence of ATP. Moreover, the current dynamic after ATP 10 and 100µM is different with and without brevenal (in A and C). Could you comment.

Response

This reviewer is absolutely correct: We compared basal currents (control, absence of ATP) +/- brevenal (unpaired t-tests) and indeed basal currents were enhanced in the presence of brevenal. This is indicated by the hash (#) symbol in the I/V curuves and is described in the text.

The reviewer raises a very interesting point. Indeed, the time dependence of the ATP-activated current is showing a continuous increase in the presence of brevenal, but decays in the absence of brevenal. Also, activation of the inward currents at negative clamp voltage is more pronounced in the presence of brevenal. Activation of TMEM16A inward currents is observed at larger intracellular Ca2+ levels {Duran, 2010 #4994}. Both findings could be either explained by a) enhanced ATP-induced Ca2+ increase in the presence of brevenal, or b) higher Ca2+ sensitivity of TMEM16A in the presence of brevenal. Because we did not find a higher ATP-induced increase in intracellular Ca2+ by brevenal (Figure 8), we conclude that brevenal enhances Ca2+ sensitivity of TMEM16A. However, because we used Fura-2 to measure global intracellular Ca2+ levels, it could still be that brevenal is enhancing Ca2+ very locally and only in close proximity to TMEM16A. This is now discussed in more detail.

    • Are you sure the holding voltage (line 417) is -100mV? What is the equilibrium potential for Cl in your experimental conditions? Please show the protocol used to record your currents.

Response

We apologize for our previous misleading description. This has now been corrected: “The cells were kept in current clamp (cc). In intervals cells were voltage clamped from -100 mV to +100 mV in steps of 20 mV. Afterwards we returned to current clamp”.  A voltage protocol has been included.

    • The inward current is clearly more affected than the outward current after brevenal. Could you compare both more precisely? For example the current amplitude at the assumed potential of -100 mV (fig 7 panel C) is above 3nA but the I/V curves in D do not show that. These experiments need clarifications.

Response

We thank this reviewer for alerting us. Indeed, brevenal induces particularly larger ATP-induced inward currents. The explanation for this is provided above. The mean I/V curve (Figure 7D) provided in the previous version of the manuscript, did not include all 14 measurements. This has now been revised. However, because in some experiments activation of the inward current was not quite as strong as in the example shown in Figure 7C, the mean values in Figure 7D are below 3 nA. The increase in delayed activation at positive clamp voltage and the pronounced activation of inward currents is now described and also discussed.

    • The results with Brevenal are not discussed in the discussion. Why?

Response

We apologize for omitting a discussion on the brevenal results, which has now been included. In the initial version of the manuscript we tried to keep the manuscript as short as possible. “The present data indicate that acute application of Eact, the activator of TMEM16A [32], acutely releases airway mucus and contracts airways in vivo. We observed that some of the animals presented with severe breathing problems after application of Eact. Although we did not observe much of an increase in intracellular Ca2+ by Eact in an earlier study [33], Eact may be able to directly increase intracellular Ca2+, e.g. by activating transient receptor potential (TRP) channels [34] and may therefore cause pulmonary effects independent of activation of TMEM16A. We therefore examined the effects of brevenal, another putative activator of TMEM16A [49]. We found in the present study that brevenal indeed enhances basal activity of TMEM16A and facilitates activation of TMEM16A through purinergic stimulation with ATP. Moreover, we found no evidence for a direct or indirect rise of intracellular Ca2+ by brevenal. This interesting compound therefore probably acts as an activator and potentiator of TMEM16A expressed endogenously in human airway epithelial cells (Figures 7 ,8). Brevenal appears to enhance Ca2+ sensitivity of TMEM16A, which is reflected by enhanced time-dependent current activation and enhanced activation of TMEM16A inward currents (Figures 7). Similar to Eact, also brevenal caused acute mucus release and bronchoconstriction in vivo, which in several animals caused severe breathing problems. The present results therefore strongly contraindicate the use of brevenal in inflammatory airway disease, unlike two recent reports which favor brevenal as a treatment in chronic respiratory disease [42,50]. In contrast to these reports, our study was done in animals with induced asthma, which leads to goblet cell metaplasia and upregulation of TMEM16A and a number of proinflammatory proteins, including SPDEF. As shown in Figure 1, uninflamed lung tissue and macrophages [51] express very little or no TMEM16A.“  

Minor :

M&M: the method describing how the cross-sectional area is determined is missing

Response

The method has now been included. We apologize for the mistake.

M&M: line 408: no inside-out patch clamp experiments have been done here. It is indicated current-density but all the data are presented only as current (nA). Please carefully check the M&M.

Response

We apologize for these mistakes It has been corrected.

Line 13 : share first authorship? no indication in the author list

Response

Thank you so much for detecting this mistake. (It was unintentionally removed by the editors)

Line 62: found in patients

Response

corrected

Line 95-97: I guess ref to Fig 2C should be indicated here. The sentence of conclusion is not relevant in this part and too premature since these data are from OVA-animals

Response

We fully agree with your criticism and therefore removed the statement,

Line 100: the activator: repetition in text

Response

Has been deleted

Line 102: Bar indicates or Bar indicate

Response

Thank you! It has been corrected.

Line 125: Bars…respectively. 2 values but 3 conditions. Please precise

Response

The method has now been included. We apologize for the mistake.

Line 153: inhibited acute…amend here

Response

It has been corrected.

Line 163: GAPDH

Response

It has been corrected.

Line 164: SPDEF should be defined above line 161 for example

Response

SPDEF is now defined.

Line 173: Figure 7? Should be Figure 6 I think

Response

Thank you for detecting the mistake. It has been corrected.

Line 176: Figure 6: do you mean 6B and C?

Response

This has now been specified: It is Figure 6B,C.

Line 192-194: add at least one ref for Brevenal here

Response

A reference has been included

Line 197-199: why adding TRAM-25? No reference here?  traces without TRAM-25.

Response

The inhibitor of Ca2+-activated KCNN4 K+ channels, TRAM-34 (100 nM), was present in all patch clamp experiments to avoid potential activation of Ca2+ activated K+ channels. This statement is now included into the figure legend and Methods.

Legend Fig E1 and E2: please indicate what mean yellow and red arrows

Response

The purpose/meaning of arrows is now included into the legend.

Legend Fig E3: tracheas

Response

This has been corrected.

Lines 225-229: sentence should be in the discussion rather than here

Response

The sentence has been removed and is now included into Discussion.

Reviewer 2 Report

The manuscript titled "Mucus Release and Airway Constriction by TMEM16A May Worsen Pathology in Cystic Fibrosis Lung Disease" is very well written. The data presented very well and conclusions are consistent with the data. There are some minor points to look out for to improve the manuscript:

  1. Results look good and significant, however, I very recommend using one- or two-way ANOVA for statistical analysis instead of t-test and specify in the methods and figures legends which method was used.
  2. It would be nice to additionally analyze the pro-inflammatory biomarkers in the bronchoalveolar fluid of mice. 

Author Response

Reviewer 2

The manuscript titled "Mucus Release and Airway Constriction by TMEM16A May Worsen Pathology in Cystic Fibrosis Lung Disease" is very well written. The data presented very well and conclusions are consistent with the data. There are some minor points to look out for to improve the manuscript:

Response

We thank this reviewer for his/her very positive statement.

  1. Results look good and significant; however, I very recommend using one- or two-way ANOVA for statistical analysis instead of t-test and specify in the methods and figures legends which method was used.

Response

Where appropriate, the results have been tested by one-way ANOVA and this test has been included. This is now specified in figure legends and Methods.

  1. It would be nice to additionally analyze the pro-inflammatory biomarkers in the bronchoalveolar fluid of mice. 

Response

Thank you for your excellent comment. In retrospective we agree that it would have been nice and truly of value to include pro-inflammatory biomarkers. In fact, in our previous study we presented evidence for a role of TMEM16A for LPS-induced release of the cytokine IL-8. {Benedetto, 2019 #8088}. We currently have a study up and running with BCi-NS1 human airway epithelial and other cell types expressing TMEM16A. In this study, cells are exposed to activators (Eact) and inhibitors (niclosamide, Ani9) and cytokine release is determined. Initial results from these studies fully confirm the role of TMEM16A for cytokine release. A separate manuscript on these results is in preparation.

Round 2

Reviewer 1 Report

Authors provided convincing replies to my comments. Thank you.